# Biodegradable Mulching Film vs. Traditional Polyethylene: Effects on Yield and Quality of San Marzano Tomato Fruits

**DOI:** 10.3390/plants12183203

**Published:** 2023-09-07

**Authors:** Ida Di Mola, Eugenio Cozzolino, Lucia Ottaiano, Riccardo Riccardi, Patrizia Spigno, Milena Petriccione, Nunzio Fiorentino, Massimo Fagnano, Mauro Mori

**Affiliations:** 1Department of Agricultural Sciences, University of Naples Federico II, 80055 Portici, Italy; ida.dimola@unina.it (I.D.M.); lucia.ottaiano@unina.it (L.O.); nunzio.fiorentino@unina.it (N.F.); fagnano@unina.it (M.F.); mori@unina.it (M.M.); 2Council for Agricultural Research and Economics (CREA)—Research Center for Cereal and Industrial Crops, 81100 Caserta, Italy; 3ARCA 2010 S.c.a.r.l., Via G. Leopardi 18, 81030 Teverola, Italy; ricc.riccardi@libero.it (R.R.); patspigno@hotmail.com (P.S.); 4Council for Agricultural Research and Economics (CREA)—Research Center for Olive, Fruits and Citrus Crops, 81100 Caserta, Italy; milena.petriccione@crea.gov.it

**Keywords:** sustainable agricultural practices, antioxidant activity, polyphenols, flavonoids, ascorbic acid, carotenoids

## Abstract

Mulching is a common practice for improving crop yield and obtaining an out-of-season production, but when made using plastic materials it can bring environmental problems due to the management and the disposal of films at the end of the cropping seasons. To increase the sustainability of this practice, recently, mulching films made with biodegradable organic materials have become more widely used. Our aim was to evaluate the effect of a biodegradable mulching film on yield and qualitative traits of the San Marzano tomato fruits over two years (2014 and 2015). Two different types of mulching were tested: (i) black biodegradable film (MB12) and (ii) black low-density polyethylene (LDPE) were compared to bare soil (BS). Both mulching films elicited a 25% increase in yield, mainly due to the significantly higher number of fruits per square meter, compared to BS. Both mulching films also elicited a 9.9% increase in total soluble solids and a 57% increase in carotenoid content, while firmness showed the highest value in BS fruits. MB12 determined the highest value of the Hunter color ratio a/b of tomato fruits, followed by LDPE, while the lowest value was recorded in BS fruits. Both mulching films elicited an increase of 9.6%, 26.0%, and 11.7% for flavonoids, polyphenols, and AsA, respectively. In 2014, the MB12 degradation started at 71 days after transplant (DAT); in 2015, at 104 DAT. Therefore, replacing polyethylene with biodegradable film would seem to be an agronomically efficient and environmentally sustainable practice.

## 1. Introduction

The global annual production of fresh tomatoes amounts to approximately 180 million tons, of which about a quarter are cultivated for the processing industry, making tomato the world’s leading vegetable for processing [1]. Of the world’s tomato cultivation, 90% is concentrated in the Northern hemisphere from April to October; the other 10% is processed in the Southern hemisphere (except for Brazil) between January and June [1]. The production is strongly concentrated in ten countries (California, China, Italy, Spain, Turkey, Iran, Portugal, Brazil, Chile, Ukraine), accounting for about 83% of the world’s yearly production (data referred to 2016–2021) [1].

According to ISTAT data [2] in Italy, the area devoted to the open-air cultivation of tomatoes for processing is 74.041 ha with a production of 5.211.512 tons, of which about 234.500 tons are produced in the Campania region on 3.884 hectares.

Today, mulching has become a common practice in the cultivation of many horticultural crops and it can play a key role in satisfying the need to make agriculture more sustainable, since, overall, it is considered a sustainable practice for saving soil and water [3]. The main goals of mulching are to limit water losses [4,5,6], improve the soil microclimate, boost temperature [7,8], and suppress weeds [9]. The linked beneficial effects of mulching are (i) crop growth stimulation; (ii) yield improvement; and (iii) increase in water and nutrient efficiency, by reducing water demand and fertilizer requirements [10]. The most used mulching film is the low-density polyethylene (LDPE) because it is low-cost, an easily processed material with high mechanical resistance, long durability, and water impermeability [11]. The consumption of LDPE mulching films increased by 35% between 2006 and 2017 worldwide [11,12]. Globally, the consumption of plastic materials has quadrupled over the past 30 years, driven by growth in emerging markets; however, consequentially, also the global plastic waste generation has more than doubled from 2000 to 2019, reaching 353 million tons, of which about two-thirds comes from plastics with lifetimes of under five years [13]. Specifically, the wide use of polyethylene in agriculture strongly impacts the agricultural environment, with the production of large amounts of plastic waste to dispose of. The total amount that is estimated to be released into the land systems is 4- to 23-fold higher than that released into marine environments [14].

The current worldwide policy (governments and companies) aims to promote the development and use of biodegradable plastics [15]. Nowadays, the market offers many types of biodegradable mulching films (BMD) [16] which are able to improve the yield of several crops with a performance close to LDPE films [10,17,18,19,20]. For example, Mendonça et al. [21] tested the effect of the different types of soil mulching (brown, green, black, silver/black, white/black and yellow/brown plastic film; rice straw mulching; no mulching and with herbicide application; no mulching and with weed hoeing; and no mulching and no weed control) on the development and yield of tomato cv. Cordillera^®^ (Feltrin, Farroupilha, Brazil) and found that the tomato plants grown on rice straw mulching reached yield similar to those on plastic films. In China, Abduwaiti et al. [22] compared the effects of PE film and biodegradable film mulching on the agronomic and economic performance of a processed tomato and they reported similar yields between plastic and biodegradable films. Contrarily, there is not much evidence about the effect of the use of biodegradable mulching film on the quality of tomato or other vegetable products. Di Mola et al. [23] compared three mulch treatments (two different biodegradable films and bare soil) and four biostimulant treatments (Ascophyllum nodosum extract; microbial biostimulant containing Trichoderma afroharzianum; a combination of both; and not treated) on tomatoes and found that both biodegradable films elicited a significant increase in lipophilic antioxidant activity and ascorbic acid content. Cozzolino et al. [24] found that a MaterBi^®^ mulching film used for melon cultivation elicited an increase in total soluble solids (13.3%), polyphenols (22.4%), flavonoids (27.2%), and antioxidant activity (24.6%) compared with LDPE films.

Therefore, considering that the results mentioned above regarding vegetable quality and the agronomic responses to biodegradable mulch can also vary depending on the cultivar, the type of biodegradable film and the conditions of cultivation, the current research was aimed at evaluating the effect of a biodegradable mulching film both on yield and qualitative traits (firmness, total soluble solids, color parameters, carotenoids, and antioxidant activity and compounds) of the San Marzano tomato fruits. Specifically, two different types of mulching were tested, (i) black biodegradable film (MB12) and (ii) black low-density polyethylene (LDPE), comparing them to bare soil (BS) over two years.

## 2. Results

### 2.1. Air and Soil Temperatures

The meteorological trends of the two years were quite different (Figure 1). The mean temperature of the whole 2014 growth period (18 April–13 October) was 22.2 °C with a maximum of 33.8 °C in the third ten-day of August and a minimum of 9.3 °C in the last ten days of April (Figure 1A). In 2015, the average temperature of the tomato growing period (21 April–24 August) was higher (24.4 °C) than that of the first year, with a maximum of 39.2 °C in the first ten days of August and a minimum of 12 °C recorded again at the end of April (Figure 1B).

For the rainfall, the differences between the two years were even larger; in fact, the total amount of rainfall was 559.3, and 194.2 mm, respectively in 2014 and 2015 (Figure 1). These differences were due partially to the longer tomato growing period in the first year, but above all to a very intense storm event recorded in the middle of June with as much as 127.8 mm concentrated in one single event, and to the unusually high precipitation of July (85.2 vs. 39.9 mm in 2015) and September (143.6 mm) (Figure 1).

All mulching films elicited an increase in the mean temperature of soil compared to bare soil during both tomato cycles (Figure 2). Particularly, in 2014, the greatest increase was recorded under the LDPE, with a mean soil temperature for the whole cycle of 21.5 °C against 20.8, and 20.7 °C for MB12, and BS, respectively (Figure 2A).

In 2015, it was warmer, and the differences between the temperatures under mulching films and bare soil were more marked, with an increase of 2.3, and 0.9 °C for LDPE, and MB12, respectively (Figure 2B).

### 2.2. Yield, Yield Components, and Firmness of Tomato Fruits

The marketable yield of tomato was statistically affected by both mulching films and years. Notably, both mulching films elicited a 25.0% increase in yield, without statistical differences between them, although the best performance was recorded for MB12 (+28.8% over BS) (Figure 3). In the second year, tomato marketable yield reached a higher value than in the first year (52.2 and 47.7 tons per hectare, respectively).

The highest yield of MB12 plants was mainly due to the higher number of fruits per square meter, even if it was not different from LDPE, and they produced an average 31.3% increase as compared to BS (Table 1). By contrast, no significant differences were found in the average weight of fruits (Table 1). As regards the main effect of year, in 2015 the number of tomato fruits per square meter was statistically higher than that of 2014 but with a lower average weight (Table 1).

Mulching films also elicited a 9.9% increase in total soluble solids of tomato fruits compared to bare soil; by contrast, firmness showed the highest value in fruits of plants grown on bare soil and the lowest in those of plants grown on LDPE (Table 1).

Finally, neither of the two parameters was affected by year (Table 1).

### 2.3. Color Parameters and Carotenoids of Tomato Fruits

Mulching films enhanced the Hunter ratio (a/b), but differently; MB12 had the best performance with the highest a/b value of tomato fruits, followed by LDPE, while the lowest value was recorded in fruits of plants grown on bare soil (Table 2).

Carotenoid content was also significantly improved by mulching films but without differences among them; on average, they elicited a 57.0% increase compared to values of fruits of plants grown on bare soil (Table 2). No effect of the year was recorded for either of the two parameters (Table 2).

### 2.4. Antioxidant Activity and Compounds of Tomato Fruits

Flavonoids, polyphenols, antioxidant activity (AA), and ascorbic acid (AsA) were statistically affected by mulching, and all, except AA, were also affected by year (Table 3).

Overall, for all the qualitative parameters, tomato fruits of plants grown on bare soil always showed the lowest values and they were statistically different from all other treatments except for AA, for which BS was not different from LDPE (Table 3).

Both mulching films elicited an increase of 9.6%, 26.0%, and 11.7% for flavonoids, polyphenols, and AsA, respectively (Table 3). As regards the antioxidant activity, only MB12 produced a 24.1% increase compared to bare soil values (Table 3).

As regards the effect of year, the highest values of polyphenols and ascorbic acid were recorded in 2015, contrarily to that observed for flavonoids.

### 2.5. Degradation of Biodegradable Mulching Film

The statistical analysis revealed that in both years, during the first 70 days after placing the mulching (DAPM), no degradation of MB12 was observed (Table 4 and Table 5). Particularly, in 2014, for all the three parameters of degradability of the MB12 film (degradation of buried and unburied film, and resistance to tearing), the degradation started at 71 days after placing the mulching and increased in the two successive sampling dates. At 162 DAPM, the resistance characteristics of the biodegradable film were almost completely compromised (the values of the three parameters ranged between 1.0 and 2.0 on a scale with a minimum rating score of 1: Table 4).

In the second year, the degradation was less marked than the first year, indeed at the last sampling, the values of the three parameters were around 5. As regards the degradation of buried film, the only differences were between the first and the last sampling, while for degradation of unburied film, the value of the last sampling was also statistically different from that of the second one and, finally, with regard to the resistance to tearing, only the first two samplings showed no difference between them (Table 5).

## 3. Discussion

Mulching is a common practice for improving crop yield and obtaining an out-of-season production (for instance, early production), but it is also an important tool for a sustainable agriculture (by maintaining a higher water content in the soil, water irrigation amounts are lower, and by suppressing weeds, weed chemical control is reduced). The most used material for mulching soil is low-density polyethylene but, in terms of environmental sustainability, in recent years, biodegradable organic materials are becoming more widely used.

In our study, we aimed at testing the suitability of mater-bi^®^-based mulching film for tomato (type San Marzano) cultivation, in terms of quantity and several quality traits of fruits.

Among the beneficial effects of mulching, there is an improvement of soil microclimate by boosting temperature [8]; also in our research, we found an increase in soil temperature in both years, but that due to LDPE was greater than that of MB12: +0.8 and +0.1 °C, and +2.3° and +0.9 °C, in 2014 and 2015, respectively, compared to bare soil. Our findings are also consistent with those of Jia et al. [25] which found that polyethylene film always elicited a greater soil temperature increase than four tested biodegradable films at all depths investigated by the authors (5, 10, 15, and 20 cm). Presumably, the differences found between the LDPE and MB12 are mainly due to the different thickness of the two films (50 vs. 12 µm), also according to Li et al. [26]. However, in spite of the reduced thickness, in both years the biodegradable film assured a good covering of soil up to the most advanced stages of the cycle. In fact, in the first year, the degradation of both buried and unburied film, and the resistance to tearing were significantly reduced starting from 71 days after transplant (DAT); in 2015, the degradation started at 104 DAT. The different behavior of the MB film in the two years could be due to the different meteorological conditions, especially as regard the rainfall. Indeed, in 2014, the total rainfall was 559.3 mm in 162 days of cycle, while in 2015 it was only 194.2 mm in 134 days. Notably, the rainfall distribution was also markedly different; in fact, in 2014, there was a storm event of almost 130 mm in mid-June and a rather rainy July (85.2 vs. 39.9 mm of July 2015).

However, as expected, mulching elicited a 25% increase in marketable yield compared to bare soil; interestingly, the increase was even 28.8% for MB12 but without significant differences from LDPE. The higher yields were mainly due to the higher number of fruits per square meter recorded for plants grown on both mulching films: +34.5% compared to the unmulched soil, while no differences were recorded for the average weight of fruits (61.9 g fruit^−1^). Our results are consistent with those of Abduwaiti and colleagues [22] who reported a 22.1% increase in tomato yield grown on mulching compared to bare soil, without differences between the polyethylene and three of the four tested biodegradable films. Jia et al. [25] also reported that mulching (polyethylene vs. four different degradable films) elicited an average increase in processing tomato yield of 21.1% compared to yield of tomato plants grown on bare soil, but in their research the PE yield was significantly higher than all the degradable films. Sekara et al. [27] compared a black polyethylene film with two MaterBi^®^ biodegradable black films (15 and 12 µm thickness) and an unmulched control used for two processing tomato cultivars, and similarly they recorded a significant increase in yield with mulching (about 10.6%) without differences between the LDPE and the 12 µm biodegradable film.

The effect of mulching, both based on traditional or biodegradable materials, on the yield of several food crops has been already investigated by many researchers; on the other hand, the effect on food quality has still not been deeply investigated. Currently, food quality has a key role in guiding consumer choices; therefore, a new challenge for farmers is to obtain high-quality products but by maintaining high yields too. Thus, the implementation of new agricultural practices should always be evaluated both as regards the quantitative and qualitative response of the crop. Notably, in our research, mulching positively affected all investigated traits of tomato fruit quality, except firmness. Firmness, in fact, was higher in fruits of plants grown on bare soil, but these results are not properly in line with the results of previous studies. Indeed, Di Mola et al. [23], on processing tomatoes grown on two biodegradable films or bare soil and treated with four different biostimulant treatments, found that mulching elicited higher values of firmness. Similar findings (lower firmness in tomatoes grown on bare soil) are also reported by several other authors [27,28]. However, it must be considered that many factors affect firmness, including environmental (light, temperature, and moisture), genetic, physiological, and cultural factors [29], but also harvest, and postharvest conditions [30]. Therefore, we hypothesize that in our experimental conditions, other factors (e.g., the tomato cultivar cultivated) could have affected firmness.

As regards the total soluble solids of tomato fruits, mulching elicited almost a 10% increase compared to bare soil; our data are consistent with those of Morra and colleagues [28] who recorded a significant increase in TSS of fruits of plants grown on biodegradable mulching film compared to bare soil (about +5.0% and 9.0% in two experimental years). Contrarily, Mendoca et al. [21], who compared six different colored plastic mulching films, a rice straw biodegradable film, and soil not covered, did not record a difference in TSS. Finally, Jia et al. [25] recorded higher values of TSS in tomatoes grown on biodegradable films and even the lowest value in the ones grown on polyethylene.

As regards the color of tomato fruits, it is a very important discriminant for product destined to both fresh and industry use. It is mainly linked to the carotenoid content; particularly, a high content of lycopene determines the red color, while high content of beta-carotene is responsible for the orange color of fruits [31]. The color of tomato is usually expressed as ratio a/b, where a* is the red component (due to lycopene) and b* is the yellow component (due to beta-carotene). Our results highlighted the highest a/b ratio in fruits of plants grown on MB12 (2.51) and the lowest in BS fruits (2.30), confirmed also by data of carotenoid content (6.69 and 4.26 mg 100 g^−1^ fresh weight, for MB12 and BS, respectively). Sekara et al. [27] reported the data of color parameters, as a* and b*, but dividing the two parameters (a/b) results in the ratio being higher in biodegradable films and lower in BS and LDPE, which is partially in line with our findings.

Finally, as regards the bioactive compounds and antioxidant activity, both mulching films significantly improved these traits of quality, except for antioxidant activity, for which only MB12 elicited a 24.1% significant increase compared to BS. Flavonoids, polyphenols, and ascorbic acid increased by 9.6%, 26.0%, and 11.7%, on the average of the two films, compared to the data from not covered soil. It is well known that tomato is overall considered as a functional and nutraceutical food, since it is an excellent source of bioactive antioxidant compounds, including phenols, vitamin C, and lycopene [32]. These bioactive compounds have several benefits on human health [33], including protection against cancer [34,35], the reduction of inflammation [36], and the maintenance of heart health by lowering total cholesterol, LDL cholesterol, triglycerides, and the risk of atherosclerosis [37]. Our results are partially in line with those of Sekara et al. [27], who found that ascorbic acid, flavonoids, and polyphenols were higher in fruits of tomato grown on the two biodegradable films, but the data collected on LDPE were not different from bare soil; instead, for antioxidant activity, the two MB equally elicited the higher values, but LDPE showed an intermediate value and was different from bare soil.

## 4. Materials and Methods

### 4.1. Experimental Setting and Crop Management

The experiment was carried out at the private farm ARCA 2010 s.r.l., located in Acerra (N 40°57′56.462″; E 14°25′50.213″; 27 m a.s.l., Naples, Italy) in open air conditions.

Two different mulching types, (i) black biodegradable film, 12 µm thickness (MB12) and (ii) black low-density polyethylene, 50 µm thickness (LDPE) were compared to bare soil (BS). All treatments were replicated three times, amounting to a total of 9 plots, distributed in randomized blocks. Each plot was 5.2 m wide and 10.0 m long.

The test was replicated for two consecutive years (2014 and 2015).

Each year, before starting the trial, the soil was sampled at 0–30 cm depth and analyzed for the main chemical and physical characteristics (Table 6).

Mulching films were hand placed on 15 April 2014 and 21 April 2015, after laying out the irrigation hoses. The transplant was made on 18 April 2014 and 22 April 2015 with a distance between the rows of 1.3 m and between the plants in the row of 0.5 m, with a density of about 15,400 plants per hectare. For the test, the chosen crop was the processing tomato Kiros, an improved ecotype of San Marzano, with bright red color and elongated fruits, and with a sweet-sour taste.

Water losses, calculated using the Hargreaves method [38], were fully restored by irrigation. No pesticide applications were required to control pathogens and pests.

Due to the high soil content in phosphorus and potassium, only nitrogen was added according to the usual farm agricultural practices.

In the first year, the harvests were made on 11 August, 3 and 24 September, and 13 October; in the second year, the harvests were made on 10 and 24 August.

### 4.2. Mulching Film Characteristics

The biodegradable film is a starch-based raw material, made of de-structured starch treated with biodegradable polyesters, and marketed with the trade name Mater-Bi^®^ by Novamont Company (Novara, Italy) [39]. This film is compostable and certified “OK Biodegradable Soil” by the Austrian certification institute TÜV, respecting the requirements of the main regulations regarding biodegradation and environmental impact (European standards: UNI EN 13432: 2002, UNI EN 14995: 200).

The traditional mulching plastic film, a low-density polyethylene, is obtained from polyethylene resin pellets, which are heated and processed into bendable sheets of plastic film. It is characterized by easy processability, great chemical resistance, high durability and flexibility [10].

### 4.3. Yield, and Yield Components of Tomato Fruits

In both years, at each harvest, the marketable fruits of each treatment were harvested on the whole plot and they were weighed in order to determine the marketable yield, ex-pressed as tons per hectare. In addition, per each treatment, tomato fruits were counted with the aim of determining the number of fruits per square meter and the average weight of the fruits.

Finally, per each treatment, a sub-sample of tomato fruits was frozen at −80 °C and, thereafter lyophilized with a lyophilizer Crist, Alpha 1–4 (Osterode, Germany) for the determination of qualitative characteristics, as reported in the Section 4.5.

### 4.4. Color Parameters, Total Soluble Solids, and Firmness Determination of Tomato Fruits

At each harvest, on a representative sample of ten tomato fruits, the two color-space parameters (a*, and b*) were determined by a Minolta CR-300 Chroma Meter (Minolta Camera Co., Ltd., Osaka, Japan) on the two opposite sides of each fruit. In particular, a* ranges between −60 (green) and +60 (red), and b* ranges between −60 (blue) and +60 (yellow), according to the Commision Internationale de l’èclairage (CIELAB). The data were reported as red/yellow ratio (a/b).

Total soluble solid (TSS) content was determined on fresh fruit juice by a digital refractometer (Sinergica Soluzioni, DBR35, Pescara, Italy); the results were expressed as °Brix.

Finally, on the two opposite sides of five fruits per replicate, firmness was measured with a digital penetrometer (T.R. Turoni srl, Forlì, Italy) with an 8 mm diameter probe, and results were expressed in kg cm^−2^.

### 4.5. Carotenoids, Antioxidant Activity and Compounds Determination of Tomato Fruits

Carotenoid content was spectrophotometrically determined at 470 nm on 1 g of fresh tomato fruits after extraction with methanol (1:10 *w*/*v*) according to the procedure reported by Wellburn [40]. Results were expressed as mg g^−1^ fresh weight (fw).

Antioxidant activity (AA) of tomato fruits was assessed by DPPH (1.1-diphenyl-2-picryl-hydrazil) method according to that reported by Brand-Williams et al. [41]. AA data were expressed as µmol Trolox equivalents (TE) g^−1^ fw, using a Trolox calibration curve.

Total ascorbic acid (TAA) was measured on frozen fresh material by a spectrophotometer according to the method detailed by Kampfenkel et al. [42], measuring the solution absorbance at 525 nm. The results were expressed as mg 100 g^–1^ fw.

The other qualitative analyses were made on samples extracted with methanol solution (80% *v*/*v*) as reported in detail by Petriccione et al. [43]. Supernatants were filtered and then used for the various assays.

Polyphenol content was determined according to the Folin–Ciocalteu method [44], and the results were expressed as mg gallic acid equivalent (GAE) per 100 g fw.

Aluminum complex formation was used to evaluate flavonoid content [45], expressing results as catechine equivalent (CAE) 100 g^−1^ of fresh weight.

### 4.6. Soil and Air Temperatures Measurements

In both years, during the whole tomato cycle, the air temperature, and the soil temperature under the different mulching films, at a 0–20 cm depth, were monitored continuously with probes (Vantage Pro2, Hayward, CA, USA, Davis Instruments).

### 4.7. Mulching Film Measurements

The resistance of the biodegradable film to atmospheric agents (mainly rainfall, wind, solar irradiance) was monitored, in both years, according to the method performed by Novamont technicians. In particular, the evaluation procedure provides a visual observation, and assigns a score ranging between 1 (worst condition) and 9 (best condition) to film. The evaluation scale is used for the following parameters: degradation of the buried and unburied film, lesions, and resistance to tearing [46]. The observations were made five and four times, in 2014 and 2015, respectively.

### 4.8. Statistical Analysis

The productive and qualitative data were analyzed with the SPSS software (SPSS version 22.0 for Windows, Chicago, IL, USA), using two-way ANOVA analysis. The data of resistance parameters of biodegradable film recorded during the two cycles were separately analyzed per each year comparing the different sampling dates. All means were separated according to the post hoc Tukey Test (significance level 0.05).

## 5. Conclusions

Our findings confirmed that mulching is a practice suitable for increasing crop yield, but interestingly we recorded a good productive performance of the MaterBi^®^-based film, that elicited yields not different from that of polyethylene. Indeed, in spite of the reduced thickness, the biodegradable film assured a good covering of soil up to the most advanced stages of the cycle, that is until when the plants were able to cover the soil. In addition, mulching also produced a significant improvement in tomato quality, with an increase in total soluble solids, bioactive compounds, and antioxidant activity, for which MB12 showed even better results than polyethylene.

A final consideration should be made about the costs related to the use of biodegradable films; in fact, by an online market research we found that a spool (100 m length × 1 m large) of biodegradable film costs EUR 28.00 against EUR 25.00 of a spool of polyethylene with the same dimension. Therefore, if one considers that to the cost of raw material must be added the costs for the removal and proper disposal of the plastic films, it is reasonable to assume that the total costs for use of biodegradable mulching are comparable or even lower than those related to the purchase of traditional plastic.

Therefore, replacing polyethylene with biodegradable film would seem to be not only an agronomically efficient but also sustainable practice, both from an environmental point of view but also economically.

Obviously, further research is needed to confirm these results with other crops.

## Figures and Tables

**Figure 1 plants-12-03203-f001:**
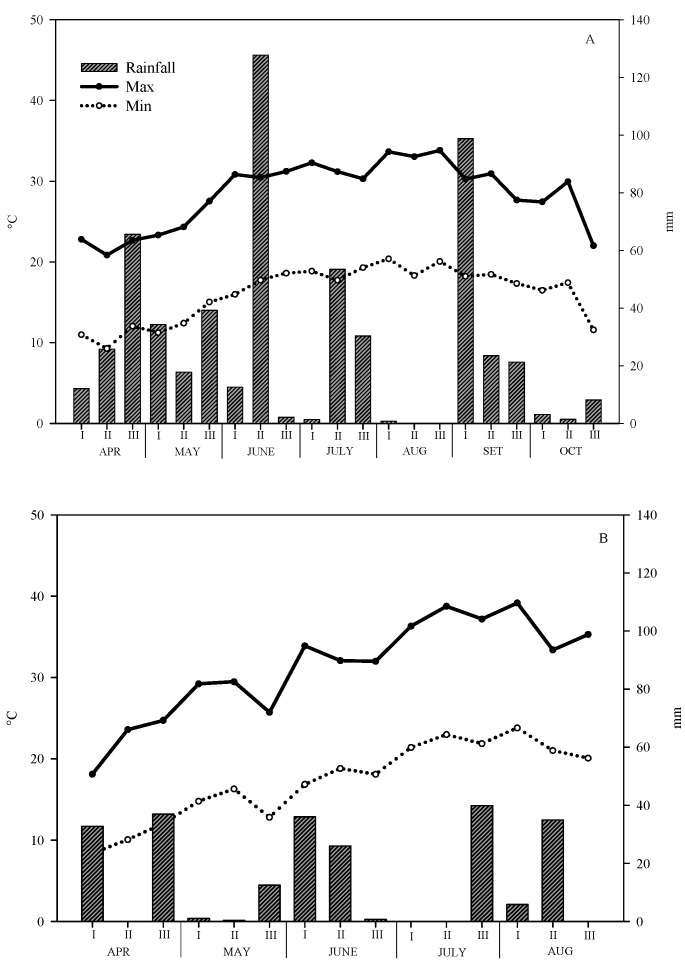
Trend of the minimum and maximum average air temperatures, and rainfall during the tomato cycle in 2014 (**A**) and 2015 (**B**).

**Figure 2 plants-12-03203-f002:**
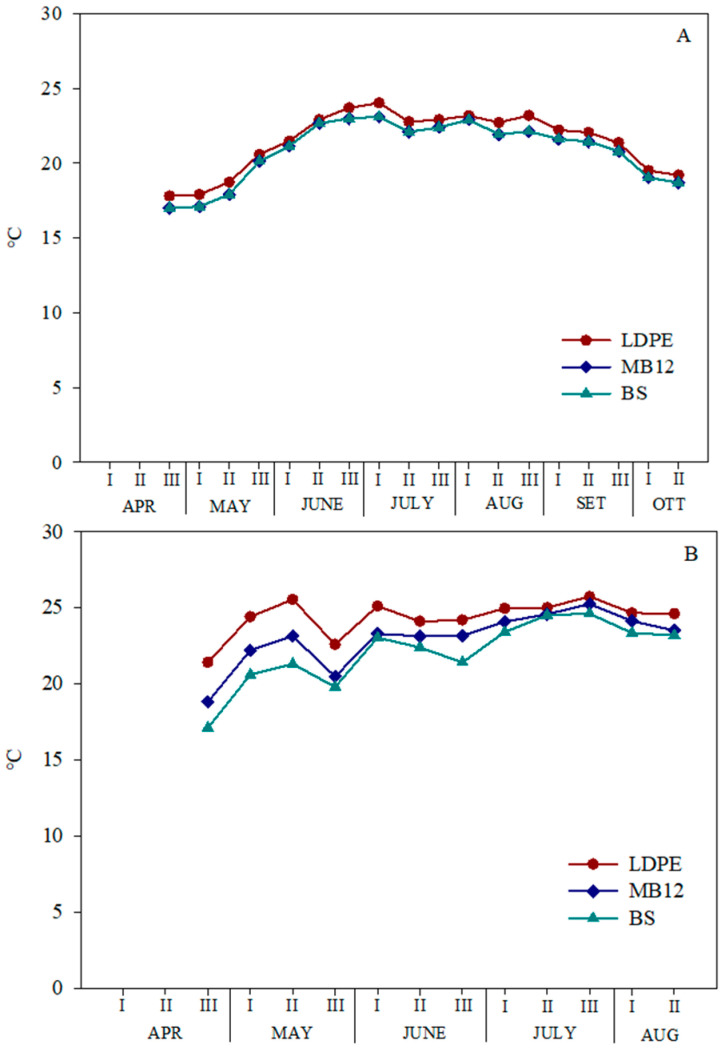
Trend of the soil average temperatures as affected by mulching (MB12: biodegradable film with 12 µm thickness; LDPE: black low-density polyethylene; BS: bare soil) during the tomato cycle in 2014 (**A**) and 2015 (**B**).

**Figure 3 plants-12-03203-f003:**
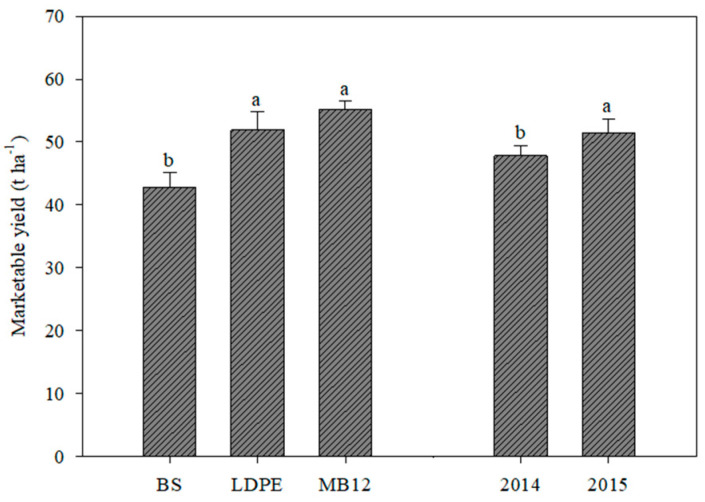
Tomato marketable yield as affected by mulching (MB12: biodegradable film 12 µm thickness; LDPE: black low-density polyethylene; BS: bare soil) and year (2014; 2015). Each column indicates the mean value of the three replicates. Vertical bars indicate standard error; different letters indicate significant differences according to Tukey test (*p* < 0.05).

**Table 1 plants-12-03203-t001:** Number per square meter, average weight, total soluble solids (TSS), and firmness of marketable tomato fruits as affected by mulching (MB12: biodegradable film 12 µm thickness; LDPE: black low-density polyethylene; BS: bare soil) and year (2014, 2015).

Treatments	Marketable Fruits		TSS	Firmness
	n m^−2^	g fruit^−1^	Brix	kg cm^−2^
**Mulching**				
BS	69.2 ± 4.5 b	62.2 ± 1.9	5.14 ± 0.06 b	0.52 ± 0.01 a
LDPE	89.5 ± 7.5 a	60.9 ± 1.3	5.72 ± 0.10 a	0.41 ± 0.01 c
MB12	92.3 ± 8.1 a	62.5 ± 3.0	5.59 ± 0.12 a	0.46 ± 0.01 b
**Years**				
2014	72.2 ± 2.2 b	66.1 ± 0.8 a	5.42 ± 0.09	0.46 ± 0.01
2015	95.2 ± 5.2 a	57.7 ± 0.7 b	5.54 ± 0.12	0.46 ± 0.01
**Significance**				
Mulching (M)	**	ns	**	**
Year (Y)	**	**	ns	ns
M × Y	ns	ns	ns	ns

ns, **: non-significant, and significant per *p* ≤ 0.01. Different letters indicate significant differences according to the Tukey test (*p* < 0.05).

**Table 2 plants-12-03203-t002:** Carotenoid content and Hunter ratio (a/b, where a* ranges from green (−60) to red (+60), and b* ranges from blue (−60) to yellow (+60)) as affected by mulching (MB12: biodegradable film 12 µm thickness; LDPE: black low-density polyethylene; BS: bare soil) and year (2014, 2015).

Treatments	Hunter Ratio a/b	Carotenoids
		mg 100 g^−1^ fw
**Mulching**		
BS	2.30 ± 0.01 c	4.26 ± 0.14 b
LDPE	2.37 ± 0.01 b	6.67 ± 0.45 a
MB12	2.51 ± 0.01 a	6.69 ± 0.47 a
**Years**		
2014	2.39 ± 0.02	5.88 ± 0.41
2015	2.40 ± 0.02	5.87 ± 0.38
**Significance**		
Mulching (M)	**	**
Year (Y)	ns	ns
M × Y	ns	ns

ns, **: non-significant, and significant per *p* ≤ 0.01. Different letters indicate significant differences according to the Tukey test (*p* < 0.05).

**Table 3 plants-12-03203-t003:** Flavonoids, polyphenols, antioxidant activity (AA), and ascorbic acid (AsA) as affected by mulching (MB12: biodegradable film 12 µm thickness; LDPE: black low-density polyethylene; BS: bare soil) and year (2014, 2015).

Treatments	Flavonoids	Polyphenols	AA	AsA
	mg CAE 100 g^−1^ fw	mg GAE 100 g^−1^ fw	µmol TE g^−1^ fw	mg 100 g^−1^ fw
**Mulching**				
BS	3.14 ± 0.08 b	110.9 ± 6.9 b	1.96 ± 0.06 b	13.0 ± 1.2 b
LDPE	3.36 ± 0.12 a	135.5 ± 7.5 a	2.18 ± 0.08 b	14.3 ± 1.0 a
MB12	3.51 ± 0.09 a	143.9 ± 9.8 a	2.69 ± 0.14 a	14.7 ± 1.2 a
**Years**				
2014	3.48 ± 0.11 a	113.0 ± 5.5 b	2.23 ± 0.10	11.6 ± 0.3 b
2015	3.20 ± 0.07 b	147.2 ± 5.5 a	2.32 ± 0.22	16.4 ± 0.5 a
**Significance**				
Mulching (M)	**	**	**	**
Year (Y)	**	**	ns	*
M × Y	ns	ns	ns	ns

ns, *, **: non-significant, and significant per *p* ≤ 0.05 and *p* ≤ 0.01. Different letters indicate significant differences according to the Tukey test (*p* < 0.05).

**Table 4 plants-12-03203-t004:** Degradation of biodegradable film (degradation of buried film and unburied film, and resistance to tearing) during the 2014 tomato cycle.

DAPM	Degradation of Buried Film	Degradation of Unburied Film	Resistance to Tearing
28	9.0 ± 0.0 a	9.0 ± 0.0 a	9.0 ± 0.0 a
43	9.0 ± 0.0 a	9.0 ± 0.0 a	9.0 ± 0.0 a
71	8.1 ± 0.1 b	8.4 ± 0.2 b	8.4 ± 0.2 b
113	7.2 ± 0.2 c	7.5 ± 0.2 c	7.2 ± 0.2 c
162	2.0 ± 0.0 d	1.0 ± 0.0 d	2.0 ± 0.0 d
**Significance**	***	***	***

DAPM: days after placing mulching; *** significant per *p* ≤ 0.001. The observations were made according to standard scoring made by Novamont technicians, based on visual observation, using a scale with a rating score from 1 to 9 (best conditions). Different letters indicate significant differences according to the Tukey test (*p* < 0.05).

**Table 5 plants-12-03203-t005:** Degradation of biodegradable film (degradation of buried film and unburied film, and resistance to tearing) during the 2015 tomato cycle.

DAPM	Degradation of Buried Film	Degradation of Unburied Film	Resistance to Tearing
55	7.3 ± 0.3 a	8.7 ± 0.3 a	7.3 ± 0.3 a
76	6.7 ± 0.3 ab	8.3 ± 0.7 a	7.3 ± 0.9 a
104	6.3 ± 0.7 ab	7.0 ± 0.0 ab	5.3 ± 0.3 b
134	5.3 ± 0.9 b	5.3 ± 0.7 b	5.0 ± 0.6 c
**Significance**	*	*	*

DAPM: days after placing mulching; * significant per *p* ≤ 0.05. The observations were made according to standard scoring made by Novamont technicians, based on visual observation, using a scale with a rating score from 1 to 9 (best conditions). Different letters indicate significant differences according to the Tukey test (*p* < 0.05).

**Table 6 plants-12-03203-t006:** Physical and chemical characteristics of the soil (0–30 cm layer) in 2014 and 2015.

Parameter	Measure Unit	2014	2015
Particle size distribution			
- Coarse sand	%	19.5	20.2
- Fine sand	%	42.1	41.2
- Silt	%	22.8	23.4
- Clay	%	15.6	15.2
Texture		Sandy loam	
N-Total *	%	0.15	0.18
P_2_O_5_ **	ppm	205.0	211.0
K_2_O ***	ppm	1500.9	1846.5
Organic matter ^#^	%	2.22	3.36
pH		7.4	7.2
Electrical conductivity ^+^	dS m^−1^	0.13	0.20

* Kjeldahl method; ** Olsen method; *** Tetraphenylborate method; ^#^ Bichromate method; ^+^ 1:5 method.

## Data Availability

The datasets generated for this study are available on request to the corresponding author.

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
