# Peer review of "Biodegradable Mulching Film vs. Traditional Polyethylene: Effects on Yield and Quality of San Marzano Tomato Fruits"

_plants, 2023, doi:10.3390/plants12183203_

Round 1
Reviewer 1 Report
This manuscript presents some interesting results about the effects of the biodegradable mulching film and the polyethylene film on the yield and quality of San Marzano tomato fruits over two years of field tests. However, it needs very significant improvement before acceptance for publication. The main problem with this work is its lack of logical organization and presentation. Some typical items are listed below.
1. It is suggested that the Introduction section should be thoroughly reorganized.
- In the end of the third paragraph of Introduction, Refs. 13 and 14 were cited to illustrate the risk of plastic pollution. However, the environmental pollution caused by polyethylene film should be described here as the previous description is “However, the large use of polyethylene in agriculture strongly impacts the agricultural environment with production of wide amounts of plastic wastes to dispose of”.
- Paragraphs 4, 5, 6, and 7 should be merged into one in the Introduction part. A paragraph should be added after this, briefly describing the main content and research methods of this work.
2. It is recommended to examine the impact of MB12 on the yield and quality of tomato in 2014 and 2015 separately, in comparison to LDPE and BS.
- Some results seem unreasonable and need to be checked carefully. For example, in Line 146, “both mulching films elicited a 25.0% increase in yield…”. And in Line 192, “Both mulching films elicited an increase of 9.6%, 26.0%, and 11.7% for flavonoids, polyphenols, and AsA, respectively.” It is apparent that the data details between the two types of films should not be exactly the same.
- In Chapters 2.2 to 2.4, a comparison of yield, yield components, firmness, color parameters, carotenoids, antioxidant activity, and compounds of tomato fruits will be presented and compared by 2014-BS, 2014-LDPE, 2014-MB12, 2015-BS, 2015-LDPE, and 2015-MB12.
Minor revision is needed.
Author Response
Dear Reviewer, thanks very much for your efforts in reviewing our paper. We corrected the manuscript according your requests, but we will upload it when we will complete also the revisions required by the other reviewers.

Reviewer 2 Report
General Summary:
The paper presents a two-season comparative study on the use of two mulching films, one biodegradable the other not (MaterBi/LDPE), with bare soil as the control bare soil), in the growing of processing tomatoes. The study‘s findings are generally consistent with other studies in that films improve yield (by ~25%). The total yield between seasons studied was different but this difference was not significant and was loosely attributed to differences in soil temperature and number plants per area. The study reports improvements in a number of qualitative attributes of the tomato including flavonoid, polyphenols and ascorbic acid but not antioxidant activity. The authors postulated that the use of MaterBi biodegradable mulch film is a feasible alternative to LDPE based on similar yield performance, product quality and guestimated agroeconomics.
Specific
· The paper has limited scientific merit given it reports on a simple assessment of a biodegradable mulch film compared to a non-biodegradable PE in tomato production. It seems to be reporting on a basic field trial rather than a scientific study. The findings on its performance are found to be consistent with previous studies and do not appear to seek or offer a great deal of any new knowledge. There is for example,
- Little evidence in the experimental design, besides trial layout, where the influence of varying input parameters (e.g. water) might be considered or determined.
- In the introduction the third listed benefit of mulching is to “increase water and nutrient efficiency”. The plants were watered during both seasons using irrigation and even though rainfall is reported the quantum of “water losses, calculated using the Hargreaves method and fully restored by irrigation” are not so there is little understanding of water use efficiency. Some comment/data regarding irrigation would be useful.
- There is little discussion on whether the physical attributes of the BDM and PE (thickness, permeability etc.) might impact outcomes.
- Comments on economic considerations have some validity but those made are reasonably shallow and unconvincing.
- The paper gives the impression that the real benefit of BDM lies in improving the quality of the product rather than in water savings or yield performance. If this is the case then perhaps this should be presented as the main purpose of the study. The way the paper is presented, this comes across as an afterthought given the yield performance of the two mulches are effectively the same.
· The English is adequate but if the paper is to be published some attention needs to be given in improving readability (see Abstract below as example).
· Replace elicict throughout the paper by produce
For example
Abstract: Mulching is a common practice for improving crop yield and obtaining an out-season production, but when using plastic mulching materials it can create bring environmental problems associated with due to the management and the disposal of films at the end of the cropping seasons. To increase the sustainability of this practice, recently, biodegradable mulching films made with biodegradable organic materials are being increasingly usedhaving a great diffusion. The Our aim of this study was to evaluate the effect of a biodegradable mulching on yield and some qualitative traits of the San Marzano tomatoes fruits over two years (2014 and 2015). Two different typologies types of mulching were tested: i) black biodegradable film (MB12) and , ii) black low-density polyethylene (LDPE) were compared to bare soil (BS). Both mulching films elicited produced a 25% increase in yield , mainly due to the significantly higher number of fruits per square meter, compared to BS. They Both mulching films also produced elicited a 9.9% increase in total soluble solids and a 57% increase in carotenoid content, while firmness showed the highest value in BS fruits. MB12 determined displayed the highest value 25 a/b of tomato fruits, followed by LDPE, while the lowest value was recorded in BS fruits. Both mulching films produced elicited an increase of 9.6%, 26.0%, and 11.7% for flavonoids, polyphenols, and AsA, respectively. In 2014, the MB12 degradation began started at 71 days after transplant (DAT); in 2015, at 104 DAT. Therefore, replacing polyethylene with biodegradable film would seem to be an agronomically efficient and environmentally sustainable. practice.
Author Response
Dear reviewer, you can find the responses to your comments in the attached file.

Reviewer 3 Report
In particular, the following necessary improvements are:
Page 1, line 25-26: “MB12 determined the highest value a/b of tomato fruits”. “Response: change in “MB12 determined the highest value of the Hunter color ratio a/b of tomato fruits”
Figure 3, “different letters indicate significant differences according to Duncan test (p < 0.05)”. Response: In the paragraph “Statistical Analysis” is reported “All means were separated according to the post-hoc Tukey Test (significance level 0.05)”. Please carefully check the manuscript contents.
Table 2, first line: modify “Hunter color parameter” with “Hunter ratio” and include it also in the table.
Table 6, Specify the soil/water ratio for the Electrical conductivity.

Author Response
Dear Reviewer, thanks very much for your efforts in reviewing our paper and for appreciating it. We corrected the manuscript according your requests, but we will upload it when we will complete also the revisions required by the other reviewers.

Reviewer 4 Report
As has been indicated in the paper, water savings with mulch can besignificant. For this reason, when a common plastic mulching is purchased
for the crop without mulching, irrigation should be separated and mulching
or not considered in the FAO formula. In this case the most adecuate will be
use sensors in the soil to irrigate with the same criteria.
Some of the results obtained may be the consequence of a greater contribution of
irrigation and not by the use of plastic mulch.
You say that no pesticide application were required to control pathogens and pest.
It is possible that they had used some preventative fungicide, such as copper
or sulfur, It is difficult to guarantee the health of the crop without using
any product. Its possible if you used sulfur, affect the differences about nbiodegradability
between the years of study.
Author Response
Dear Reviewer, thanks very much for your efforts in reviewing our paper. You can find our responses to your considerations in the attached file. We will upload the revised version of manuscript when we will complete the revisions required by the other reviewers.

Round 2
Reviewer 1 Report
I recommend publishing this paper.
Minor editing is required.